# Adapting the marine stewardship council's risk-based framework to assess the impact of towed bottom fishing gear on blue carbon habitats

Kate Morris[1]*, Graham Epstein[2,3], Michel J. Kaiser[1], Joanne Porter[4], Andrew F. Johnson[1,5]

1 The Lyell Centre, Heriot-Watt University, Edinburgh, Scotland, United Kingdom, 2 Centre for Ecology and Conservation, University of Exeter, Penryn Campus, Cornwall, United Kingdom, 3 University of Victoria, Victoria, British Columbia, Canada, 4 International Centre for Island Technology, Heriot-Watt University Orkney, Stromness, Orkney, Scotland, United Kingdom, 5 MarFishEco Fisheries Consultants Ltd, Edinburgh, Scotland, United Kingdom

* kjm2001@hw.ac.uk

**Data Availability Statement:** Fishing pressure data is available from ICES - Shapefile datasets are available at ICES (2018a): https://doi.org/10.17895/

## Abstract

Wild capture fisheries are of economic and social importance, providing a primary source of protein to people globally. There is a broad research base on the environmental impacts of fishing gears and processing methods yet, the impact on the global $CO_2$ budget is less well studied. Evaluating the risk that wild capture fisheries pose to ecosystem health is vital to sustainably managing fishing practices to meet increasing global nutritional needs and reverse declines in marine biodiversity. At the same time meeting net-zero ambitions by reducing direct and indirect GHG emissions is vital. Ecological risk assessments, trait-based assessments, and vulnerability assessments have long supported fisheries management systems globally but do not yet provide any representation regarding the impacts that fishing gears have on the ability of the habitat to capture and store carbon. Considering the importance of accessibility and transparency in approaches necessary for fisheries sustainability certifications, this paper describes a method to integrate habitat carbon capacity attributes into the Marine Stewardship Council (MSC) Consequence and Spatial Analysis (CSA) framework. Applying the CSA carbon extension developed herein produces different CSA risk scores compared to the MSC CSA that does not account for carbon. This has potential consequences for certification schemes as carbon becomes more important in the fisheries sustainability conversation. The CSA carbon extension tool developed here is an important first step in incorporating carbon indicators into evaluations of fisheries that consider fishery carbon impacts.

## 1. Introduction

Wild capture fisheries are of global economic and social importance [1] providing 84.4 million tonnes of commercial seafood and were the primary source of protein for 3.3 billion people in

ices.data.4686 Habitat data can be requested from the UK Joint Nature Conservation Committee (JNCC) - https://jncc.gov.uk/our-work/marine-habitat-data-product-eunis-level-3-combined-map/ All calculations and supporting references are provided in the Supporting Information files.

**Funding:** Kate Morris was funded by the James Watt Scholarship provided by Heriot-Watt University. Andrew F Johnson was funded by Heriot-Watt University and MarFishEco Fisheries Consultants Ltd. Graham Epstein was funded by the Barclays Ocean Climate Impact grant. The funders had no role in study design, data collection and analysis, decision to publish, or preparation of the manuscript.

**Competing interests:** The authors have declared that no competing interests exist.

**Abbreviations:** $CO_2$, Carbon Dioxide; CSA, Consequence Spatial Analysis; EEZ, Economic Exclusion Zone; EUNIS, European Nature Information System; FCP, Fisheries Certification Process; GHG, Green House Gas; ICES, International Council for Exploration of the Sea; IPCC, International Panel for Climate Change; MBCG, Mobile Bottom Contacting Gear; MMO, Marine Management Organisation; MSC, Marine Stewardship Council; NPP, Net Primary Production; OC, Organic Carbon; RBF, Risk Based Framework; SAR, Swept Area Ratio; SubsurfSAR, $\geq 2$ cm penetration depth of the gear components; UK, United Kingdom; VMS, Vessel Monitoring System.

2018 [2]. Towed fishing gears that make contact with sea floors, such as trawls and dredges, have a wide range of ecological impacts [3–5]. These impacts include the disturbance of benthic ecosystem function, bycatch, alterations to food web structures and modification of the seafloor stucture and topography. A broad research base exists for the ecological and physical impacts of towed bottom contact fishing gears [5–7], including the removal of habitat [8], and the production and processing methods involved in wild capture fisheries are understood to contribute to the global $CO_2$ budget [9]. However, the science surrounding the impacts of bottom fishing on carbon stores and carbon emissions remains a relatively poorly studied area, other than for studies that make the link between fishery fuel combustion and carbon emissions [10, 11].

The drive to net-zero carbon [12] to meet International Panel for Climate Change (IPCC) emission targets [13], and increased efforts to mitigate the impacts of climate change have focused attention on aquatic carbon storage [14–17]. Marine habitats with high levels of carbon storage capacity are now widely considered to be a useful nature-based solution in mitigating the effects of climate change [16]. Considering the wide-reaching impacts and spatial extent of bottom towed fishing gears [18], the link between seafloor disturbance and carbon storage and emissions has become a focal area of fishery impact research [19]. Efforts to map and/or manage organic carbon stores in marine systems [11, 20, 21], and investigations surrounding the processes surrounding sediment resuspension and remineralisation of carbon [22, 23] are all adding to this important body of literature.

The benthic impacts of towed fishing gears has been highlighted as a potentially significant impact on marine carbon stores and carbon storage processes and potentially $CO_2$ emissions to the atmosphere [24]. However, the research linking benthic carbon disturbance to atmosperic changes in $CO_2$ emissions is uncertain, while modelling and global extrapolations currently rely on assumptions that are not well supported by evidence [19, 25]. Despite these uncertainties, there remains an urgent need to design accessible methods that can be used to classify the risk that different fishing activities pose to benthic carbon stores, while acknowledging the aforementioned evidence gaps [26].

Ecological risk assessments [4, 27, 28], trait-based assessments [29, 30], and vulnerability assessments [31] have been used to support many fisheries management systems globally by evaluating the magnitude of damage caused by anthropogenic disturbance events such as bottom fishing (for example, see [32]). These assessment methods have been built on our understanding of benthic community life-history traits [4, 6, 33] but do not yet provide any representation regarding the impacts that fishing gears have on the ability of an impacted habitat to capture and store carbon. Guidance on how to incorporate carbon considerations into existing fisheries management measures is thus absent from the available carbon literature.

Considering the importance of accessibility and transparency in approaches necessary for fisheries sustainability certifications, the aim of this paper is to develop and test a method to integrate habitat carbon capacity attributes into the Marine Stewardship Council's (MSC) Risk Based Framework (RBF) Consequence and Spatial Analysis (CSA) [34]. The resultant method is referred to as the CSA carbon extension and is applied to a southern North Sea as a case study example.

## 2. Method

The Marine Stewardship Council's (MSC) Consequence and Spatial Analysis (CSA) [34], originally adapted from Williams *et al*, 2011 [27] and Hobday et al, 1968 [35], involves a qualitative evaluation of habitat-productivity attributes; gear-habitat interaction attributes, and; spatial attributes (Annex 1: MSC RBF CSA scoring attributes [34]. Each distinct habitat within a

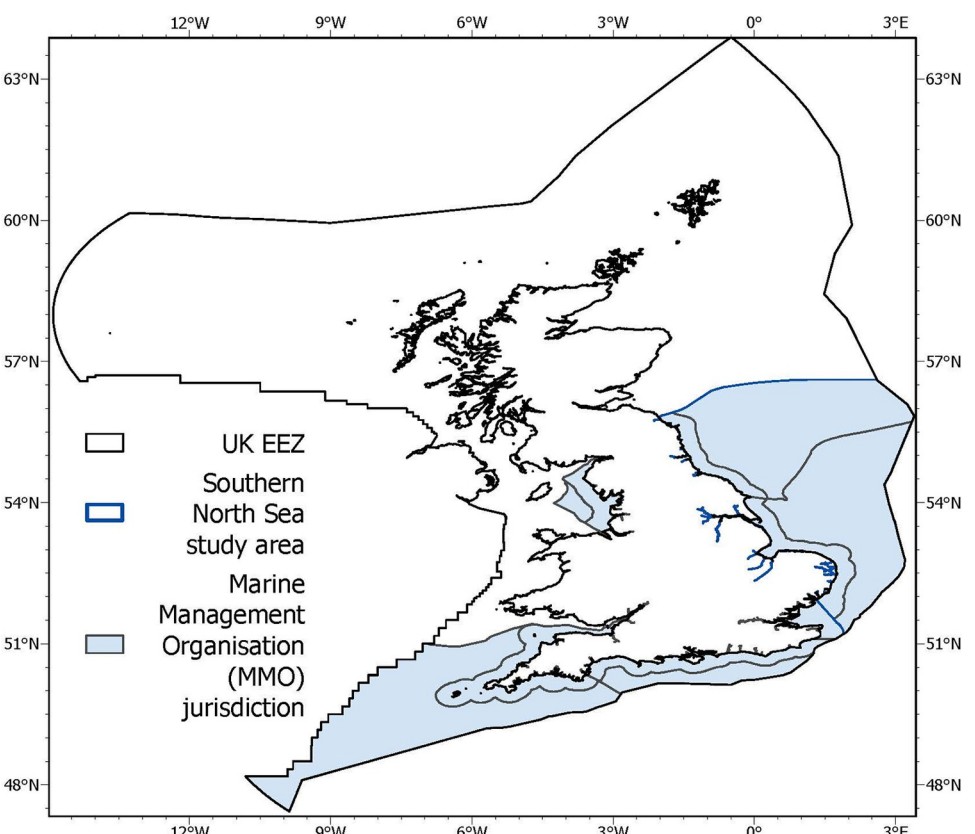

**Fig 1. Area under evaluation in the Southern North Sea case study, set in the context of the UK economic exclusion zone and marine management organisation jurisdiction boundaries.**

fishery region is defined and scored as a separate scoring element, and all scores, under an MSC CSA assessment, are calculated using the MSC RBF scoring workbook [36] and Fisheries Certification Process (FCP) v2.2 instructions and guidance [34]. Each attribute is assigned a score which is then collectively converted into a consequence score and spatial score. The consequence score describes the sensitivity of the habitat to natural and anthropogenic disturbance, whilst the spatial score denotes the fishery-specific footprint. The consequence and spatial scores are in turn converted into an overall CSA risk score for the fishery under evaluation.

The steps applied in incorporating carbon attributes into the existing MSC CSA and developing the CSA carbon extension are described in further detail in the proceeding methods sections. The CSA carbon extension has been developed for application to fisheries operating in the United Kingdom (UK) Economic Exclusion Zone (EEZ) but is tested in a southern North Sea case study Fig 1), which falls under the fisheries management jurisdiction of the Marine Management Organisation (MMO).

## 2.1 Defining assessment scope

The first step in any ecological risk assessment methodology is to establish the spatial extent of the fishery under assessment, the fishing gear used, and the habitats impacted. Within the MSC CSA and CSA carbon extension (CCSA), each distinct habitat type (known as a scoring element) must be individually scored. In some assessments, it may also be appropriate to

acknowledge environmental or fishing pressure differences within the same habitat type when distributed across large spatial scales by creating additional scoring elements. This was not done in the present assessment, which is focused on the development of a CCSA than its application. The southern North Sea case study area was selected to test the CCSA because it generally has habitat types which are representative of habitats found across the entire UK EEZ. Habitats were defined using the JNCC habitat map [37] to the third level of the European Nature Information System (EUNIS) habitat classification system. Where habitats were classified with mixed EUNIS level 3 categories, polygons were reclassified following Annex 2: Table 5 in S1 File and scored as separate scoring elements. For developing the CSA carbon extension rocky habitats act as a gauge for macroalgae distribution as rock provides suitable substrate for macroalgae growth. The fishing gears under assessment in the context of the southern North Sea case study are defined in Annex 3: Table 6 in S1 File.

All geospatial analyses were undertaken using ArcGIS Pro and all statistics were undertaken using R version X64 4.1.1.

## 2.2 Spatial scoring attributes

A spatial score denotes the fishery-specific footprint within the CSA. The spatial attributes assessed under the CSA carbon extension are swept-area ratio (SAR), spatial overlap between habitat type and fishing gear, and fishing gear footprint. The MSC CSA can be conducted using qualitative information for such spatial attributes. However, the CSA carbon extension developed herein uses quantitative information to assess the SAR attribute (fishing effort data) and spatial overlap attribute (mapped habitat extent within the managed area) to improve the accuracy of the spatial attributes input.

**2.2.1 SAR attribute.** The CSA carbon extension strengthens the MSC CSA methodology by replacing the encounter-ability attribute (see: Annex 4: Table 4 in S1 File, and MSC FCP V2.2 PF7.5.7 [34]), which qualitatively estimates the likelihood that a fishing gear will encounter the habitat, with a SAR attribute. Each SAR value is synonymous with the seabed area impacted by Mobile Bottom Contacting Gear (MBCG) and is calculated using vessel track data from Vessel Monitoring Systems (VMS), the vessel size, and gear used.

A SAR value of 0 means that the c-square (unique spatial identifiers with a grid size 0.05˚) grid area has had no recorded fishing activity [38]. A SAR value of 1 means that the entire c-square has been swept once (within the data given calendar year) by MBCG. SAR layers available through the International Council for Exploration of the Sea (ICES) from 2015 to 2017 [38] were used in the development of the CSA carbon extension and clipped to the UK EEZ boundary layer (Annex 4: Fig 5 in S1 File). SAR data is provided for <2cm sediment depth (SurfaceSAR) and ≥2cm (SubsurfaceSAR), for this analysis SubsurfaceSAR was used to align with penetration depths of MBCG. Earlier years of SAR data (2009–14) were excluded from this study so shifting fishing patterns did not bias the assessment of habitats or areas currently impacted by fishing. Data in any c-squares within the UK EEZ which contained undisclosed SAR values, were removed.

Under the MSC CSA, categorical encounter-ability scores are awarded by using Table PF17 [34] reproduced in Annex 4: Table 7 in S1 File). This was modified for the CSA carbon extension by using the SAR data and dividing the distribution of SAR values for c-squares within the UK EEZ into six equal intervals (Annex 4: Fig 6 in S1 File). Six intervals were calculated to match the current MSC CSA scoring brackets. The southern North sea Case study assesses MBCG, if an assessment were to be conducted focusing on gears that do not interact the seabed, the lowest SAR attribute score of 0.5 would be awarded.

**2.2.2 Spatial overlap and gear footprint attributes.** Assessment of the spatial overlap and gear footprint attributes broadly follows the MSC CSA process detailed in FCP v2.2.

AnnexPF7 [34]. The spatial overlap attribute is assessed as the percentage of the habitat range which falls directly within the fished area under assessment, compared to the habitat's entire range within the relevant management bodies jurisdiction (MSC FCP V2.2. PF7.5.4). The southern North Sea case study falls within the MMO jurisdictional boundary (Fig 1). Scoring the gear footprint attribute uses MSC FCP v2.2 Table PF16, which dictates a score of three (largest spatial footprint) be awarded to all demersal trawl and dredge gear.

## 2.3 Consequence scoring attributes

The consequence score describes the sensitivity of the habitat to natural and anthropogenic disturbance. The CSA carbon extension builds on attributes that are already scored under MSC CSA by accounting for additional habitat productivity attributes that indicate the ability of a habitat type to capture and store carbon. By understanding the natural distribution of OC stock, and the other associated carbon habitat characteristics, a determination can be made on where MBCG poses the greatest risk to the OC.

The consequence scoring attributes assessed under the CSA carbon extension developed herein fall under two categories, habitat productivity, and gear-habitat interaction. The habitat productivity attributes scored herein, are the regeneration of biota, natural disturbance, organic carbon (OC) stock, and OC accumulation. OC stock is defined as the average OC measure held within both sediment and macrophyte biomass components of a particular habitat type. OC accumulation is defined as the average OC accumulating in the sediment and macrophyte biomass over any given year. Gear-habitat interaction attributes are the removability of biota, removability of substratum, substratum hardness, substratum ruggedness, and seabed slope.

**2.3.1 Habitat productivity scoring attributes.** *2.3.1.1 Regeneration of biota (flora and fauna) attribute*. The regeneration of biota attribute is scored using MSC FCP v2.2 Table PF12 [34] and data from the available literature on estimated regeneration time following an anthropogenic disturbance event for each habitat type. Habitats with longer regeneration times (equal to or greater than ten years) are awarded three—the highest CSA score [39]. Habitats with regeneration times estimated to be longer than a year, but less than ten years are awarded a score of two, and habitats with regeneration times of less than a year scored one. In the southern North Sea case study, habitats are predominantly sedimentary and characterised by infauna bioturbators [40]. Biota in these habitats is likely to be displaced by MBCG but the mortality rate is expected to be low and recovery is estimated to be less than annual if left undisturbed [5]. These habitats are therefore awarded a score of one in both the MSC CSA and the CSA carbon extension. Rocky habitats, occupied by kelp canopies, when disturbed by trawls may take two to three years to recover the small understorey kelp plants and therefore awarded a score of two in both the MSC CSA and the CSA carbon extension [39, 41]. Saltmarsh and seagrass display the longest recovery times, estimated to be more than decadal [39, 41]. Saltmarsh and seagrass are awarded the highest score of three in the MSC CSA and CSA carbon extension.

*2.3.1.2 Natural disturbance attribute*. The natural disturbance attribute is scored using MSC FCP v2.2 Table PF13 [34] and relevant data from available literature on benthic sheer stress. For the southern North Sea case study natural disturbance was scored using data from Wilson *et al*, 2018 [42]. MSC FCP v2.2 Table PF13 [34] dictates that littoral habitats, defined herein as intertidal and shallow subtidal habitats, experience a higher degree of natural disturbance awarded a score of one, compared to sublittoral habitats which are deeper and therefore generally less disturbed, were awarded a score of two. The MSC scoring outcome for Natural disturbance is verified by the Wilson *et al*, 2018 [42] study which provides estimates for monthly

natural disturbance rates for the UK EEZ. The Wilson *et al*, 2018 [42] study generally supports the scores awarded within the MSC CSA and CSA carbon extension. The study highlights a difference in the rate of natural disturbance between the southern and northern parts of the southern North Sea case study area, with the southern half experiencing higher rates of disturbance in both littoral and sublittoral habitats. Habitat extents were reviewed alongside the Wilson *et al*, 2018 [42] data to determine the significance of this difference in scoring the natural disturbance attribute for each scoring element. Only the scores for biogenic reefs were changed from a two to one because the habitat extent fell entirely within the southern part of the North Sea case study area, which is estimated to experience higher rates of natural disturbance.

*2.3.1.3 Organic carbon attributes.* Incorporating the chosen carbon attributes into the CSA carbon extension draws on information from a carbon stock and accumulation synthesis report from the Centre for Environment, Fisheries and Aquaculture Science [43]. This review compiled over 500 records, from 114 publications, to establish an estimate for OC stock ($gCm^{-2}$) and the associated OC accumulation rate ($gCm^{-2}yr^{-1}$) for relevant marine habitats in English waters, and the wider UK EEZ. The review is evaluated below, alongside other literature sources, and where possible carbon attributes are incorporated into the CSA carbon extension for each UK marine habitat type. For the North Sea case study, the value assigned to each carbon attribute (stock or accumulation) per habitat was taken as an average across the entire UK EEZ.

*2.3.1.4 Organic carbon stock attribute.* OC stock is integrated into the CSA carbon extension as a single attribute made of two separate components, sediment OC stock and, macrophyte OC stock.

For the development of the CSA carbon extension and its application to the southern North Sea case study, only habitats with OC stock estimates could be included in the CSA carbon extension (Table 1). Absent from the available literature are OC stock and accumulation estimates for biogenic reefs and both littoral and sublittoral coarse and mixed sediments [14, 44]. Littoral sediments contain higher concentrations of OC than sublittoral sediments, particularly in estuaries and fjords [20]. Despite the potentially significant OC stocks contained within littoral sediments, it has not been studied to the same extent as sublittoral sediments [14]. From

**Table 1. The CSA carbon extension scores for the Organic Carbon (OC) stock ($kgCm^{2}$) attribute and OC accumulation ($kgCm^{-2}yr^{-1}$) attributes.**

| Scoring element | Habitat classification | Average OC Stock ($kgCm^{-2}$) | Average OC accumulation ($kgCm^{-2}yr^{-1}$) |
|---|---|---|---|
| n.a | Unknown low carbon capacity habitat | 0.00 | 0.00 |
| 1 | Littoral rock and other hard substrata | 0.51 | 0.340 |
| 2 | Littoral coarse sediment | n.d | n.d |
| 3 | Littoral sand, muddy sand | 6.50 | 0.045 |
| 4 | Littoral mud | 19.90 | 0.084 |
| 5 | mixed sediments | n.d | n.d |
| 6 | Coastal saltmarshes and saline reedbeds | 27.11 | 0.263 |
| 7 | Littoral sediments dominated by aquatic angiosperms | 21.33 | 0.360 |
| 8 | Sublittoral coarse sediment | n.d | n.d |
| 9 | Sublittoral sand | 1.70 | 0.030 |
| 10 | Sublittoral mud | 5.55 | 0.044 |
| 11 | Sublittoral mixed sediments | n.d | n.d |
| 12 | Biogenic reefs | n.d | n.d |

Applicable to a 1m sediment depth profile (where relevant) and the UK Exclusive Economic Zone. Full references and estimates are shown in Annex 5—Table*s* 8 (OC Stock) and 9 (OC accumulation) in S1 File. Average OC accumulation includes macrophyte net primary production where relevant. n.d = no data available.

the available literature, only OC stock (gC/m$^2$) estimates for littoral sand and mud sediments are available (Table 1 and Annex 5: Table 8 in S1 File). For sublittoral sediments, the OC stock across the North Atlantic shelf and UK EEZ has been modelled to a 0.1m depth, by several different teams [20, 21, 45–47]. Parker *et al*, 2021 [43] standardised all OC stock estimates for the UK EEZ by extrapolating estimates to a 1m sediment depth profile, assuming equal distribution of OC (Annex 5: Table 8 in S1 File).

In the southern North Sea, macrophyte habitats include kelp and other macroalgal beds, saltmarshes, and seagrass. The OC stored within the plant material and the immediate surrounding sediments are omitted from previous littoral and sublittoral OC sediment stock and accumulation mapping studies [20, 21]. However, studies have shown that in the UK, macroalgal beds [48], saltmarshes [49], and seagrasses [50] play an important role in the oceanic carbon cycle [14]. The existing macrophyte biomass stock is therefore included in calculating the average OC stock and accumulation values for the different habitat types within the UK EEZ (Table 1 and Annex 5: Table 8 in S1 File).

*2.3.1.5 Organic carbon accumulation.* Accounting for the OC stock in isolation within the CSA carbon extension would not be appropriate because, on its own, the OC stock attribute does not fully indicate the impact on OC net or storage processes. For example, areas of high OC stock may have comparatively low carbon accumulation rates. To account for this here, we use the term carbon accumulation to estimate annual net primary production in macrophytes and sediment carbon accumulation rates where relevant for each habitat [14]. By including macrophyte biomass production in the calculations of average OC accumulation, there is a risk of double counting OC as the proportion of plant detritus that is exported to sediments over relevant time-periods is poorly known. OC production estimates assume that 4–9% of the plant material produced is transported to seabed sediments as detritus [51]. The ideal adjustment would be to run the CSA carbon extension where OC accumulation estimates are based on an increased and decreased OC detritus load to understand differences in the risk assessment outcomes for each habitat type and determine which detritus transport % version to use. However, because detritus transport estimates are based on complex ecological models and require verification by in-situ measurements it is not possible within the scope of this assessment to generate new estimates for detritus transport.

Like the OC stock attribute, for the development of the CSA carbon extension and its application to the southern North Sea case study, only habitats with OC accumulation estimates could be included in the CSA carbon extension (Table 1). Furthermore, extreme estimations for OC stock and accumulation may bias the attribute scores assigned to each habitat. Saltmarsh is considered to be a habitat with higher carbon capacity, but at the other end of the scale, a dummy value of 0.00 (kgCm$^{-2}$ and kgCm$^{-2}$yr$^{-1}$) was included as a lower carbon capacity habitat.

*2.3.1.6 Scoring OC stock and accumulation attributes.* To calculate the OC stock and accumulation attribute scores, the OC estimates for stock and accumulation were normalised to a 1–3 continuous scoring range. The MSC CSA scoring used for other habitat-productivity attributes is discrete, 1, 2, or 3. This was not adopted for the OC stock or accumulation attributes scores because a continuous scale will return greater accuracy in predicting CSA risk scores (Fig 2).

Saltmarsh habitat has the highest OC stock and accumulation estimates, however, due it's spatial distribution, it's likely these carbon rich habitats are not going to be targeted by fishermen. By including Saltmarsh, the higher OC stock and accumulation estimates may lower the CSA risk scores produced for other habitat types which spatially are more at risk to fishing. To investigate the effect saltmarsh has on the overall CSA scores, OC stock and accumulation attribute scores were recalculated without Saltmarsh and the full CSA carbon extension run on

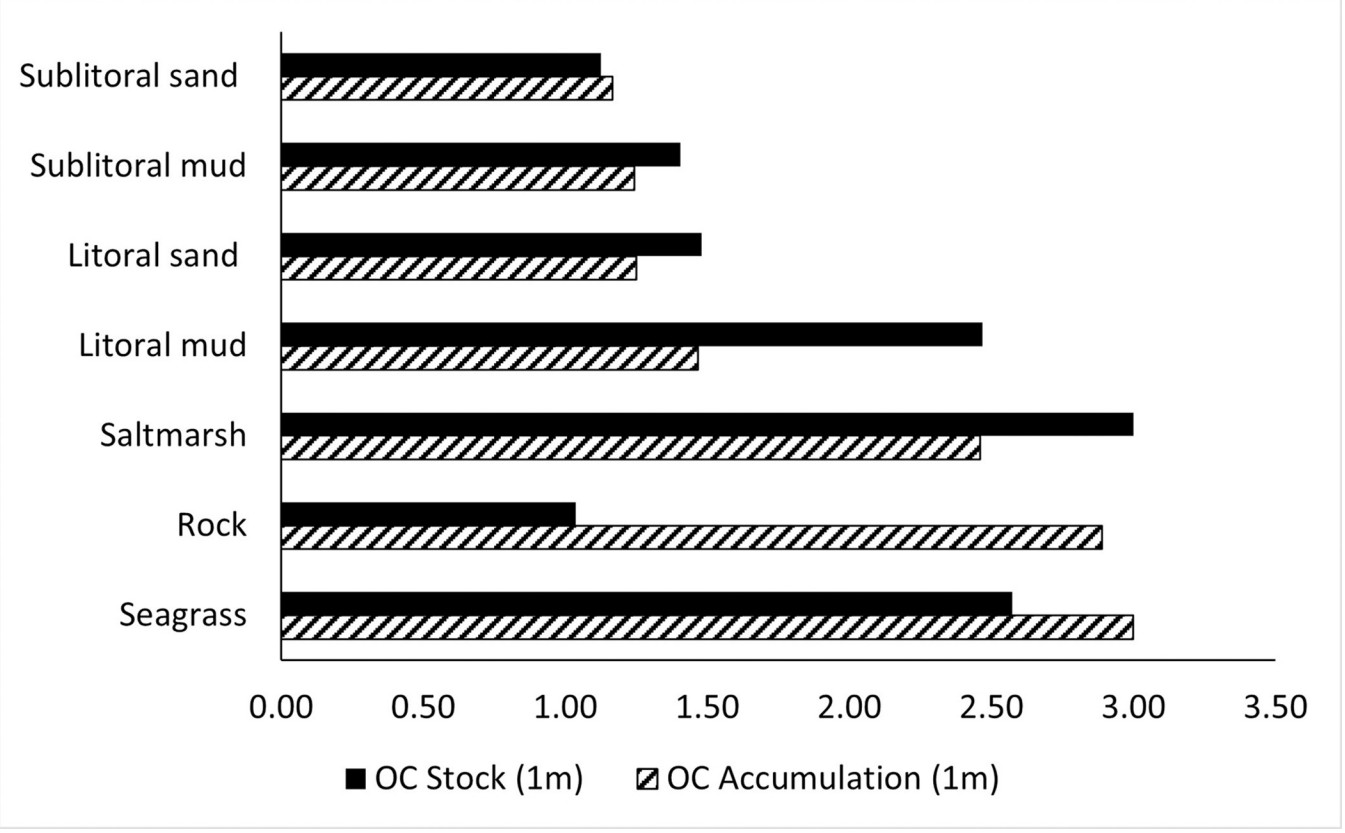

**Fig 2. Organic carbon stock (Orange) and accumulation (Blue) attribute scores for the different habitat types (scoring elements) assessed in the CSA carbon extension.** 1- lower attribute score, 3-higher attribute score.

all habitat types (excluding saltmarsh). Risk scores were compared to those produced when saltmarsh is included in the development of the CSA carbon extension. When Saltmarsh is not included the overall CSA risk score increases by 0.05–1.48% for carbon rich habitats and remains largely the same for habitats lower in carbon (S2 File). It would be beneficial to include saltmarsh in the framework because saltmarsh is at risk to other resource extractive methods and if reclaimed by the sea through sea level rise, may become economically viable fishing grounds. Saltmarsh is precautionarily included in the final development of the CSA carbon extension, (Fig 2).

**2.3.2 Gear-habitat interaction scoring attributes.** *2.3.2.1 Removability of biota and sub-stratum attributes.* The removability attributes assess the likelihood that the biota and substra-tum encountered by MBCG are moved or damaged. The attribute score assigned is therefore gear-specific for the different habitat types. For example, it is generally assumed that hand col-lection methods have less of an impact on both biota and substratum encountered compared to dredges or demersal trawls [52]. The removability attributes are scored using MSC FCP v2.2 Table PF14 [34]. As Table PF14 of the MSC CSA assigns scores based on gear type, southern North Sea case study scores were conservatively assigned using the dredge gear type qualifier for all MBCG—i.e. given a score of three across all scoring elements for removability of biota because dredges are known to exert the most damage [6]. For the removability of substratum attribute all habitats except for rocky habitats, in the MSC CSA and CSA carbon extension, were awarded the highest attribute score of three. Rocky habitats were awarded an attribute score of one because MBCG in these environments is unlikely to damage the substratum.

*2.3.2.2 Substratum hardness*, *ruggedness and*, *seabed slope attributes.* The substratum hardness, substratum ruggedness and seabed slope attributes are scored using MSC FCP v2.2 Table PF15 [34]. In the MSC methods, substratum hardness, ruggedness and seabed slope attributes aim to capture the appropriateness of the habitat to fishing activities, whilst considering the gear-specific impact. Using MSC FCP v2.2 Table PF15 [34] each scoring element is categorised and scored by the gear type under assessment (see S2 File for scores assigned to each scoring element). The substratum hardness categories are: Hard (igneous, sedimentary, or heavily consolidated rock types); Soft (lightly consolidated, weathered, or biogenic) and; Sediments (unconsolidated). The substratum ruggedness categories are: High relief (>1m, high outcrop, or rugged surface structure (cracks, crevices, overhangs, large boulder, rock walls); Low relief (<0.1m, rough surface structure, rubble, small boulders, rock edges, subcrop, or low outcrop) and; Flat (simple surface structure, mounds, undulations, ripples, current rippled, wave rippled or irregular. The seabed slope categories are: Low degree (<1, plains in coastal margin, inner or outer shelf or mid-slope, terraces in mid-slope, rocky banks, fringing reefs in coastal margin, inner or outer shelf or upper or mid-slope); Medium degree (1–10, terraces in outer shelf or upper slope) and; High degree (>10, Canyons in outer shelf, or upper or mid-slope, seamounts bioherms in coastal margin, inner shelf or upper or mid-slope). For consistency in the southern North Sea case study, scores were awarded using the dredge gear type qualifier which is the most destructive to benthos of the fishing gears listed by the MSC.

In addition to following FCP v2.2 Table PF15 [34] instructions, for the southern North Sea case study assessment of the seabed slope attribute utilised the Emodnet-Bathymery portal [53] and the depth contours used to validate the scores assigned. The seabed slope in the southern North Sea case study area is a shallow plain with a slight elevation visible around the Doggerbank which is included entirely within the habitat category, sublittoral sand. The Doggerbank slopes are estimated to be 5˚ on the south side and 0.1˚ on the north [54]. The southbank slopes of the Doggerbank comprise a small portion of the overall habitat extent, therefore the attribute score for sublittoral sand did not change because of this sloped area.

## 3. Results

### 3.1 Calculating CSA carbon extension risk score

We incorporated two carbon attributes into the MSC CSA to form the CSA carbon extension: OC stock and OC accumulation (Table 2).

Once the attribute scores have been assigned, the overall CSA risk score is calculated in three steps.

Step 1: calculate the Consequence score:

$$\text{Consequence score} = \frac{\Sigma(\text{Habitat prod. attribute scores}*2) + \Sigma\text{Gear : Habitat interaction attribute scores}}{\text{total number of attributes}}$$

Step 2: calculate the Spatial score:

$$\text{Spatial score} = \text{Geometric mean (Spatial attribute scores)}$$

Step 3: calculate the CSA risk score using outputs generates in steps 1 and 2:

$$\text{CSA risk score} = 0.5\sqrt{(\sqrt{\text{Consequence score}} + \sqrt{\text{Spatial score}})}$$

The overall CSA risk score, for both the MSC CSA and CSA carbon extension, is the euclidean distance between the consequence and spatial scores if it were to be plotted on an a two-

**Table 2. Description of the calculations used to determine the overall Consequence Spatial Analysis (CSA) risk score for each scoring element in the MSC CSA and the CSA carbon extension.**

| | | | |
|---|---|---|---|
| Regeneration of biota | **Habitat-productivity attributes** | **Consequence score** | **CSA risk score** |
| Natural disturbance | | | |
| Organic Carbon stock* | | | |
| Organic Carbon accumulation * | | | |
| Removability of biota | **Gear-Habitat interaction attributes** | | |
| Removability of substratum | | | |
| Seabed slope | | | |
| Substratum ruggedness | | | |
| Substratum hardness | | | |
| Gear footprint | **Spatial attributes** | **Spatial score** | |
| Spatial overlap | | | |
| Swept Area Ratio* | | | |

Asterisks indicate new attributes included in the MSC CSA to form the CSA carbon extension.

axis plot [27] and is presented as a value ranging from 1.18 (low risk)– 4.24 (high risk). In the MSC CSA and CSA carbon extension, all attributes, except for the habitat-productivity attributes, carry equal weighting in the calculation of the final CSA risk score. The equation for calculating the final CSA risk score is taken from the MSC methodology, which has gone through vigorous stakeholder consultations over many years. Although it may be useful to run the CSA carbon extension with attributes weighted based on their importance to an overall risk score, when considering the science surrounding fisheries' impacts on marine carbon is still in its infancy, we suggest that subjectively assigning weightings to the different attributes is not a useful exercise at this time based on the current state of knowledge.

## 3.2 Southern North Sea case study CSA risk scores

By including carbon attributes into the CSA carbon extension, we demonstrate that habitats with high OC stock and accumulation estimates have higher CSA risk scores than those produced in a MSC CSA (Fig 3). Habitats with higher OC stock and / or accumulation estimates score on average 3% higher CSA risk scores in the CSA carbon extension compared to the MSC CSA. The greatest increase in risk score of 3.67% was observed in seagrass followed by saltmarsh (+2.64%) and rock (+2.73%). Conversely, habitats with comparatively lower OC stock and / or accumulation estimates produced lower CSA risk scores in the CSA carbon extension, compared to the MSC CSA. The greatest decrease in a CSA risk score was observed in sublittoral mud (-4.39%), followed by sublittoral sand (-4.00%) and littoral sand (-3.08%). Littoral mud experienced little to no change (0.4 increase).

When applying the MSC CSA and CSA carbon extension methodology the order of risk changes slightly. The top and lowest three CSA scores for habitats in both CSA and CSA carbon extension are the same. Sublittoral sand, saltmarsh, and seagrass are the highest scoring (Fig 3). Littoral sand, rock and sublittoral mud all have the lowest CSA risk score, with littoral sand falling 0.0063 below rock. Further observed changes, are under the CSA carbon extension, seagrass is scored higher, exchanging places with littoral mud, and rock and littoral sand score the same. These changes in risk are more clearly visible when the MSC CSA and CSA carbon extension risk scores are mapped per habitat type for the entire Southern North Sea case study (Fig 4A and 4B), particularly for the lower CSA carbon extension score associated with sublittoral sand. Mapping risk on a smaller scale inshore, the Wash area of the Southern

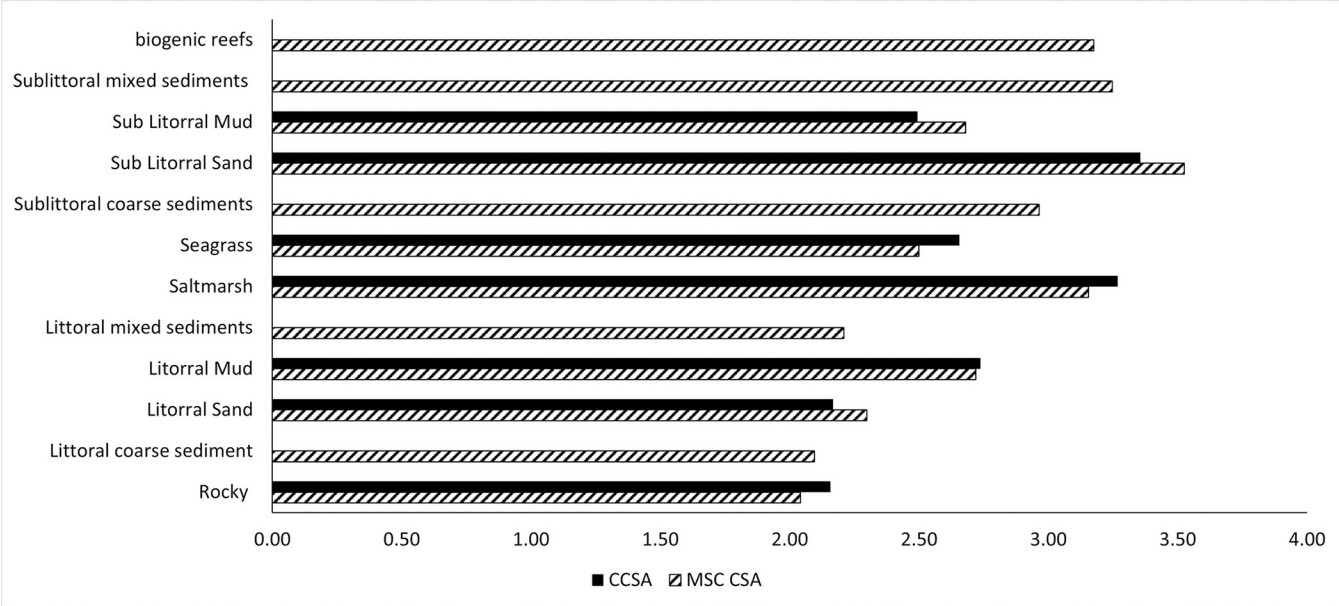

**Fig 3. Comparison of CSA risk scores from a MSC CSA assessment (solid colour) and a CSA carbon extension (pattern colour).** Scores are presented for the twelve different habitat types (scoring elements) in the southern North Sea case study area in order of risk: highest risk—lowest risk. The CSA Carbon extension could not be applied to sublittoral mixed sediments, biogenic reefs, littoral mixed sediments or littoral coarse sediments due to lack of carbon attribute estimates (see Tables 10 and 11 in S1 File).

North Sea case study provides a great diversity of inshore habitats to map the CSA risk score change (Fig 4C and 4D). In Fig 4C and 4D the decrease in sublittoral and littoral sand CSA risk scores is observed as the mapped colour scale darkens.

Finally, across the southern North Sea case study, the predominant habitat type is sublittoral sand which scores one of the highest MSC CSA score and the highest CSA Carbon extension score (Fig 3). Sublittoral sand scores one of the highest Spatial scores (2.82), only closely topped by biogenic reefs and sublittoral mixed sediment (spatial score = 2.47), which also have the highest MSC CSA risk scores. This result demonstrates that fishing pressure (SAR) is a key driver of the overall CSA risk score.

## 4. Discussion

The importance and interest of carbon storage in marine systems is growing rapidly with the drive towards net zero [12, 55]. The CSA carbon extension (an adaptation of the MSC CSA method) described herein may be used to help inform the sustainable, climate conscious, development of UK fisheries. It draws on published OC stock and accumulation estimates calculated for habitats within the UK EEZ and is tested using a southern North Sea case study. The results presented highlight which marine habitats are most at risk to MBCG in the southern North Sea and demonstrates that the inclusion of carbon in the MSC risk-based framework would offer different risk scores that are used to define sustainable practices through fisheries certification.

To identify risk, or an acceptable level of risk to any habitat is no easy task, especially in a complex marine ecosystem. Yet, tools are needed to help provide the best assessment of marine habitats and inform fisheries managers, scientists, and policy makers, on how we use the marine space. The CSA carbon extension developed herein is the first attempt made to incorporate carbon indicators into an extant risk-based framework [34]. It does not specify an

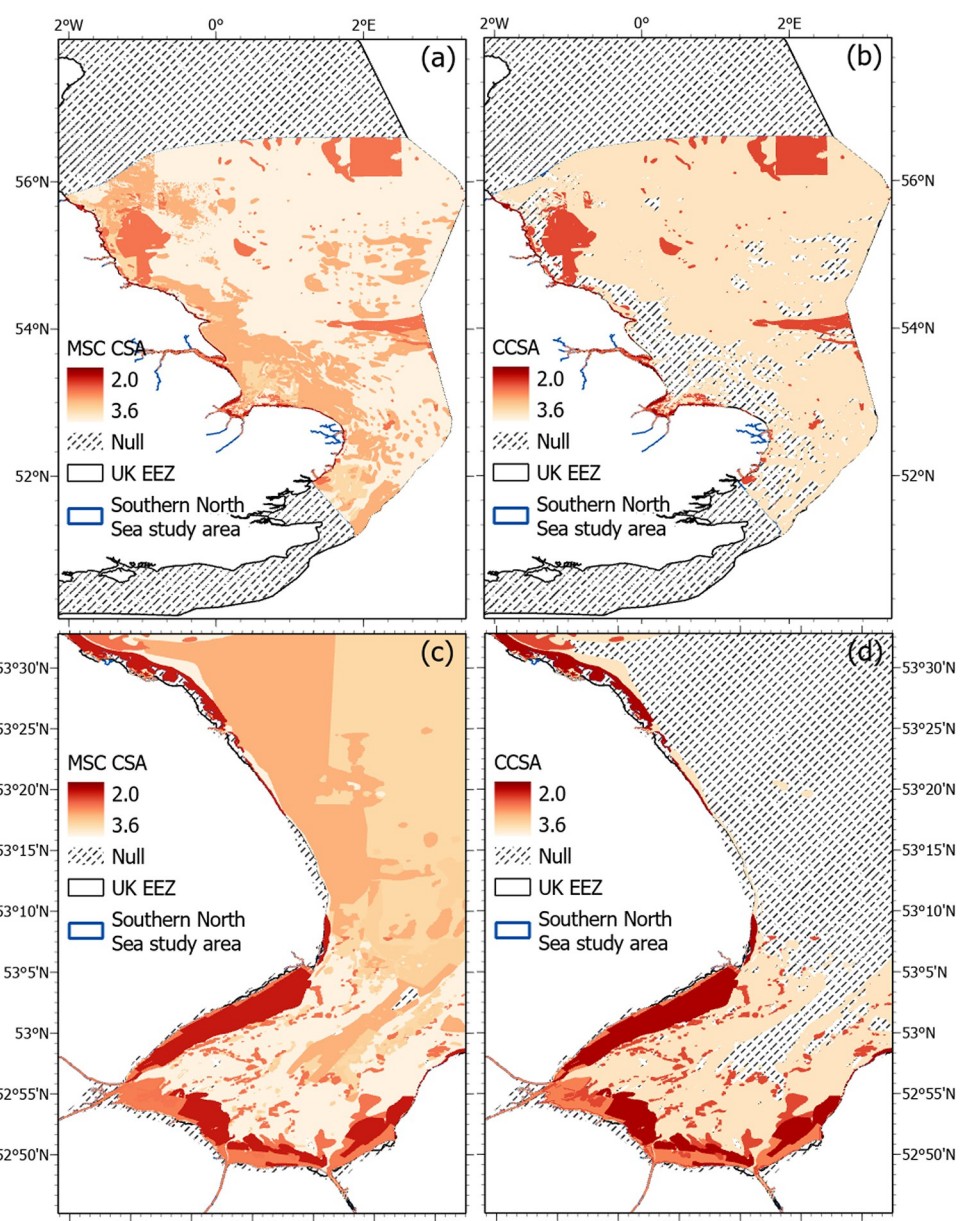

**Fig 4.** CSA risk scores across different habitat types in the southern North Sea case study area under the MSC CSA (a) and the CSA carbon extension (CCSA) (b). An area of inshore habitats around the Wash Bay are shown in panels (c) for the MSC CSA and (d) for the CSA carbon extension. The habitat range estimates use the JNCC EUNIS level three habitat map [37], which extrapolates sampled areas to predict the entire range of a habitat within defined polygons. The resulting CSA scores are assigned each habitat polygon to create these maps. The grey <null> areas are where habitats lack OC stock or accumulation estimates and therefore excluded from the CSA carbon extension assessment. The white area, outside of the southern North Sea case study area, is not assessed and therefore no CSA scores are presented.

acceptable level of risk but provides a working solution to assess marine habitats with a consideration of carbon storage in marine habitats. The Southern North Sea case study was used to test the CSA Carbon extension and to understand difference in CSA results. The CSA results produced are: specific to the fishery under assessment; scale dependent and; uniform across the habitats range within the assessment area.

Applying the CSA carbon extension to vessels >12m, deploying MBCG, in the southern North Sea produces different CSA risk scores compared to the MSC CSA that does not account for carbon. Where the OC stock and / or accumulation estimates for a given habitat are comparatively low compared to the average across the UK EEZ, the overall CSA risk score decreases. This is because the consequence score is calculated as an average of the habitat-productivity attributes and gear-habitat interaction attributes, so low scoring OC stock and accumulation estimates lower the overall consequence score.

The CSA risk scores are also influenced by the Spatial scores, with habitats experiencing greater fishing pressure being assessed as more at risk. The highest Spatial scoring habitats (for vessels within the case study area with the highest fishing efforts) are sublittoral sand and littoral mud respectively. Sublittoral sand is the dominant habitat type across the southern North Sea resulting in the highest spatial overlap and SAR scores. However, sublittoral sand stores comparatively lower amounts of OC and scores one of the lowest consequence scores of the six habitats assessed in the case study area. Therefore, when interpreting the CSA risk score and developing management actions, the importance of the consequence score and spatial score should not be considered in isolation.

The risk scores derived from the CSA carbon extension are best supported by the Smeaton and et al, 2022 [56] study on the quantity and quality of OC in different UK marine sediments. MBCGs disturb and potentially degrade natural OC stocks in marine habitats by mixing and resuspending the sediment, which may increase the rate of remineralisation both in the seabed and water column [45, 57]. Smeaton and *et al*, 2022 [56] showed that inshore sediments generally have higher reactivity and therefore following disturbance remineralisation is likely to be higher when compared to sediments found further offshore [56]. Even if this stored carbon is remineralised it is currently unknown how much will reach the atmosphere as $CO_2$ [45]. However, it is possible that by increasing the local concentration of inorganic carbon in water column, MBCG may impact the flux of carbon from the ocean to the atmosphere [58]. Over time this may impede the ability of the ocean to sequester excess anthropogenic $CO_2$. This, however, needs further investigation [45, 59].

The reliability of the CSA carbon extension risk scores depends on the accuracy of the evidence used to determine attribute scores per habitat type. The OC stock and accumulation estimates used to develop the CSA carbon extension are based on a wide variety of studies, spanning several decades, all with differing objectives, terminologies and sample depths. OC stock and accumulation estimates, are also only available for a proportion of the habitats relevant to the southern North Sea case study, those for biogenic reefs, littoral and sublittoral coarse and mixed sediments are absent from the available literature. As extreme estimations for OC stock and accumulation may bias the attribute scores assigned to each habitat, the unkown OC stock and accumulation estimates for these habitats may change the overall CSA Carbon extension risk scores. Future CSA carbon extension assessments may also consider additional or alternative carbon attributes such as OC burial and lability where there is currently a paucity of research. OC burial quantifies the amount of OC accumulating which is subsequently buried in sediments for longer geological timescales whereas lability describes the reactivity of a compound. Most of the OC that accumulates in sublittoral sediments has already reacted with the surrounding environment, degrading the OC content [60]. Therefore, the vulnerability of blue carbon habitats, with less labile OC, such as, sublittoral sediments, to fishing gear disturbance may be lower despite high OC stock estimates [56].

In addition to improving consistent reporting of OC attributes, any future CSA carbon extension assessment should make use of more accurate habitat and fishhing pressure data. Habitat range estimates use the JNCC EUNIS habitat map [37], which extrapolates sampled areas to predict the entire range of a habitat. The UK shelf is well sampled [61] but even a good

habitat map cannot fully capture the complexity and diversity of benthic community structures which can vary greatly over small distances. For carbon rich habitats, who's climate mitigation potential has only recently been recognized, these habitats may have been underrepresented in targeted habitat surveys in previous years. Or assumptions are made in the absence of accurate mapping, for example, this study assumes 100% coverage of macroalgae on all favorable rocky habitats, which is likely an overestimation of macroalgae coverage. The MSC CSA and CCSA methodology relies on habitat maps being available, a greater level of detail will produce more accurate CSA results. The spatial resolution and format of the habitat maps compared to other environmental variables must also be considered. In this study the habitat ranges were adjusted (Table 5 in S1 File) by combining more granular, mixed, habitat descriptions to a single third EUNIS level which provided a larger habitat area to assess. An increase in habitat range may equate to an increase in risk score [62]. Similarly, a decrease in habitat range will result in a decrease in the overall risk score, or result in some habitats not being assessed, therefore using more granular habitats maps with smaller ranges and the associated more comprehensive attribute knowledge would improve the accuracy of the CSA assessment [62]. The CSA results are also uniform across a habitats entire assessed range, and therefore cannot account for spatial variation within a habitat type. An alternative method to apportion fishing pressure data across a habitats range, would be to split a single habitat type into smaller scoring elements based on the SAR distribution. Finally, The fishing pressure data used in this assessment comes from ICES OSPAR data, which is from vessels fitted with VMS, typically over 12m in length, and assumes equal pressure across a c-square. These larger vessels are unlikely to be fishing close to shore. Therefore the fishing pressure data used herein are likely not the most accurate for predicting risk in inshore and coastal zones. Without corroboration from primary data collection from inshore vessels, the spatial and overall CSA Carbon extension risk scores are considered conservative.

The CSA carbon extension produced herein should be applied to more case studies and results validated by comparing risk scores with results from in-situ experimental studies evaluating the impact that fishing gears may have on carbon stored. Furthermore, as better spatial and habitat attribute data becomes available the CSA carbon extension can be strengthened and re-run to provide more accurate results. Future versions of the CSA carbon extension could consider the risks of other extractive activities such as deep-sea mining or coastal dredging. Carbon attributes developed herein, are specific to UK EEZ marine habitats. Literature reviews of carbon stock and accumulation rates for marine habitats in other countries would facilitate any replication of this work to develop region-specific CSA carbon extensions. Where data for a region-specific case is not available, we suggest using proxies or next best estimates so as not to stall the development and use of this method, if error / inaccuracies associated with such data use is readily acknowledged.

Finally, the CSA carbon extension presented here is an assessment of risk and not a GHG emission abatement tool, because the link between the seabed and sea-air flux is still largely unknown [63]. As this knolwedge gap improves and the drive for net-zero continues, the CSA carbon extension should become integral to existing seafood certification systems, such as the MSC's capture fisheires standard and any potential future standalone carbon-in-fisheries standards.

## Supporting information

**S1 File.**
(DOCX)

**S2 File.**
(XLSX)

## Acknowledgments

KM was funded by the James Watt Scholarship provided by HWU. AFJ was funded by HWU and MarFishEco Fisheries Consultants Ltd. GE was funded by the Barclays Ocean Climate Impact grant. We would like to thank Samira Anand and Sarah Stephenson for their work collating information that formed the background of this paper.

## Author Contributions

**Conceptualization:** Kate Morris, Andrew F. Johnson.

**Data curation:** Kate Morris.

**Formal analysis:** Kate Morris, Graham Epstein.

**Methodology:** Kate Morris.

**Supervision:** Michel J. Kaiser, Joanne Porter, Andrew F. Johnson.

**Writing – original draft:** Kate Morris.

**Writing – review & editing:** Kate Morris, Graham Epstein, Michel J. Kaiser, Joanne Porter, Andrew F. Johnson.

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
