## [Decision Letter · Decision Letter 0]

28 Mar 2023

PONE-D-23-06266Adapting the Marine Stewardship Council’s Risk-Based Framework to assess the impact of towed bottom fishing gear on blue carbon habitatsPLOS ONE

Dear Dr. Morris,

Thank you for submitting your manuscript to PLOS ONE. After careful consideration, we feel that it has merit but does not fully meet PLOS ONE’s publication criteria as it currently stands. Therefore, we invite you to submit a revised version of the manuscript that discusses the points raised during the review process.

in particular the limitations of the CSA

We look forward to receiving your revised manuscript.

Kind regards,

Judi Hewitt

Academic Editor

PLOS ONE

Journal Requirements:

3. We note that Figure (1, 4 and 5) in your submission contain copyrighted images. All PLOS content is published under the Creative Commons Attribution License (CC BY 4.0), which means that the manuscript, images, and Supporting Information files will be freely available online, and any third party is permitted to access, download, copy, distribute, and use these materials in any way, even commercially, with proper attribution. For more information, see our copyright guidelines: http://journals.plos.org/plosone/s/licenses-and-copyright.

1. You may seek permission from the original copyright holder of Figure (1, 4 and 5) to publish the content specifically under the CC BY 4.0 license. 

Reviewers' comments:

Reviewer's Responses to Questions

**Comments to the Author**

1. Is the manuscript technically sound, and do the data support the conclusions?

Reviewer #1: Yes

Reviewer #2: Yes

2. Has the statistical analysis been performed appropriately and rigorously? 

Reviewer #1: Yes

Reviewer #2: I Don't Know

3. Have the authors made all data underlying the findings in their manuscript fully available?

Reviewer #1: Yes

Reviewer #2: Yes

4. Is the manuscript presented in an intelligible fashion and written in standard English?

Reviewer #1: Yes

Reviewer #2: Yes

5. Review Comments to the Author

Reviewer #1: Morris and colleagues describe a method for integrating habitat carbon capacity attributes into a Consequence and Spatial Analysis (CSA) framework. Applying this novel CSA produces higher CSA risk scores for some habitats compared to the status quo method reflecting a more complete overview of the risk associated with fishing impacting on habitat productivity.

The integration of habitat carbon capacity attributes to the CSA framework is a useful and topical addition. The manuscript is very well written and well referenced. The Introduction in particular is logical, flows well and steps the reader through the rationale and the current literature.

My comments on the manuscript on the whole are relatively minor. However, I am skeptical of the CSA approach given some of the scoring, and the way it is mapped spatially. I accept that for risk assessments / models that simplifications and assumptions are inevitable, and I acknowledge that CSA has been developed by others (I’m not tasked with reviewing the CSA here). However, I think that acknowledgement of some of the (what I perceive to be key) limitations and how it affects the authors' interpretations would be useful.

Specifically:

If I’ve understood correctly, CSA scores are calculated for the habitat type across the study area. That is, you have a risk score from 1 (low risk) to 4 (high risk) for each habitat type (irrespective of the extent of spatial distribution, noting your SAR can vary spatially). Your SCORE IS SCALE DEPENDENT? That is, if your assessment was undertaken at different spatial scales (let’s say a quarter of your study area), you might get different risk scores for each habitat type, but this isn’t considered here. If my interpretation is correct then I think you need to caveat this.

Secondly, the authors map risk spatially, but noting that there is no variation spatially by habitat type in the current assessment. I think this needs to be made very clear, and you could also point to other assessment / methods that do allow for spatial risk to vary by habitat. For example, the RBS method which you cite (not a risk assessment per se but has been used in risk assessments) does vary spatially within habitat types because the benthic status is calculated from the distribution of bottom fishing swept area.

Thirdly, I’m somewhat confused that Sublittoral mixed sediments and sublittoral sands have a higher consequence score than most (including rocky reefs and Biogenic reefs!). This is irrespective of the SAR. I understand that you haven’t come up with these estimates, and that these have been extensively developed by others, but I think it would be worthwhile making it clear somewhere why the scores are what they are and why they make sense, because intuitively as an ecologist they don’t to me. This doesn’t need to be long, e.g., you could include a short justification / driver for each habitat type in the appendix next to the scores?

Finally, the use of EUNIS at the broad scale is acknowledged as a limitation in the discussion. But I think more of this needs to be brought out. For example, highlighting the sheer number of possible biological communities that can be associated with each of the level 3 habitats. I.e., the risk, must be different between reefs with soft corals vs those with mixed hydroid / bryozoan turf with encrusting pomatoceros worms (a very common habitat in some parts of the UK). I think that the spatial variation in these biological communities would greatly influence the spatial representation of risk and should be explicitly acknowledged in the discussion. I think the authors can again point to methods may address these limitations in the future and how carbon attributes will vary between these communities (i.e. point to future work, no change to approach here).

Further minor comments:

Introduction: Check heading format / numbering for publication.

L 173: Can you provide some examples of the aforementioned examples applied to real world fisheries management? E.g., see the joint Australian / New Zealand Bottom Fisheries Impact Assessment: https://www.sprfmo.int/science/bottom-fishing/ But it would be good to provide a couple more examples given you state risk assessments have supported “many fisheries management systems”

Figure 1. Consider (but not essential) relabelling the graticules to indicate cardinal points rather than – or + signs. I.e., 10°W (rather than -10); you can also remove the North arrow and scale bar if you have graticules on the map. Both of these points are personal preference and I’ll leave it to the authors / editor to decide.

L 212-214: it is not clear where you are getting level 4 EUNIS classification from? Where are the maps for these? My search of reference 36 only shows EUNIS level 3. Can you include the map in the supplementary materials or a link to a GIS map?

General comment about EUNIS habitats: These are very broadscale. i.e., 0 – 25m for littoral kelp dominated habitats is not very realistic for many parts of the UK; just because there is rock at 20m does not mean there is L digitata (suggest this could be better estimated with some kind of estimate of turbidity / PAR to the seafloor estimate if available – or pointed to in the discussion as a future direction). The overestimate of kelp is acknowledged in the discussion, but see my comments about more emphasis on how scores may change with other biological communities accounted for (and ideally how their spatial distribution can also affect spatial distribution of risk).

L242: Please provide the grid size of c-square.

L246: formatting of reference

L294: Looking at Fig 6 in appendix, you can you get SAR > 1 (at least some areas are fished more than once per year). If so, how do you account for this in the risk assessment? I.e. should the CSA score be 1.5 or 2 for SAR > 1? Whether such a change would result in a different risk outcome is the key question. If not, then ok to leave as 1, but if this particular score has a high influence on the final risk assessment, then worth considering where SAR > 1 should have a different risk score. Either way, I think this should be considered in greater detail the discussion.

L296: are there risks associated with cumulative impacts resulting in a shift in dominant biota. I.e. the kelp does not recover?

L384: I think the authors need to be careful here. By normalising the scores they are effectively saying that CSA score (risk) MUST go from low – high and that scores are relative to each other. I don’t have an issue with the relative to each other part of the assumption. What I’d be most concerned about is stating that there is a low CSA score for some habitats when they are actually all at moderate – high risk (i.e., there are no “low” carbon capacity attributes). I think this possible limitation needs to be acknowledged and the authors need to justify why they think this normalisation (rather than a scoring based on evidence / expert knowledge) is the best approach and does not produce incorrect CSA scores.

L421 – I don’t understand how or why CSA aims to capture how “economically advantageous the habitat is to fishing activities” and how that feeds into a risk assessment? Suggest clarifying.

Section 2.1 (first section of results) – I suggest this is moved to the methods.

Fig 4. It might be the low resolution of the figures in the draft manuscript, but suggest that you could make Null values clearer (either diff colour, or with hashing, or something).

L 601: consider providing examples of methods which would allow SAR to be calculated inshore (e.g., using Fisher knowledge, or IFCA monitoring):

Turner, R.A., Polunin, N.V.C., and Stead, S.M. (2015). Mapping inshore fisheries: Comparing observed and perceived distributions of pot fishing activity in Northumberland. Marine Policy 51, 173-181.

Szostek, C.L., Murray, L.G., Bell, E., and Kaiser, M.J. (2017). Filling the gap: Using fishers’ knowledge to map the extent and intensity of fishing activity. Marine Environmental Research 129, 329-346.

Reviewer #2: The study is a useful extension of the MSCs risk assessment framework. It would be helpful for MSC assessment improvement and in giving a more fulsome picture of certified fishery sustainability if this new benthic carbon disturbance risk analysis methodology becomes a new plug-in to MSC assessments in future.

Further to that I have minor editorial comments that you’ll no doubt pick up in your final edit.

6. PLOS authors have the option to publish the peer review history of their article (what does this mean?). If published, this will include your full peer review and any attached files.

Reviewer #1: No

Reviewer #2: **Yes: **Katherine Short

---

## [Author Response · Author response to Decision Letter 0]

9 Jun 2023

Thank-you to both reviewers for their time in giving this manuscript a review. Their comments were valuable and comprehensive.

---

## [Editor Report · Decision Letter 1]

28 Jun 2023

Adapting the Marine Stewardship Council’s Risk-Based Framework to assess the impact of towed bottom fishing gear on blue carbon habitats

PONE-D-23-06266R1

Dear Dr. Morris,

We’re pleased to inform you that your manuscript has been judged scientifically suitable for publication and will be formally accepted for publication once it meets all outstanding technical requirements.

Kind regards,

Judi Hewitt

Academic Editor

PLOS ONE
---

## [Editor Report · Acceptance letter]

4 Jul 2023

PONE-D-23-06266R1 

Adapting  the  Marine  Stewardship  Council’s  Risk-Based  Framework  to  assess  the  impact  of  towed  bottom  fishing  gear  on  blue  carbon  habitats  

Dear Dr. Morris:

I'm pleased to inform you that your manuscript has been deemed suitable for publication in PLOS ONE. Congratulations! Your manuscript is now with our production department. 

Kind regards, 

on behalf of

Dr. Judi Hewitt 

Academic Editor

PLOS ONE